# Occurrence and Abundance of an Apex Predator and a Sympatric Mesopredator in Rural Areas of the Coastal Range of Southern Chile

**Fernando García-Solís** [1,2,*], **Jaime R. Rau** [2] and **Edwin J. Niklitschek** [3]

1   Programa de Doctorado en Ciencias, Mención Conservación y Manejo de Recursos Naturales, Universidad de Los Lagos, Avda, Fuchslocher 1305, Osorno 5290000, Chile
2   Laboratorio de Ecología, Departamento de Ciencias Biológicas y Biodiversidad, Universidad de Los Lagos, Osorno, Castilla 933, Osorno 5290000, Chile; jrau@ulagos.cl
3   Centro i-mar, Universidad de Los Lagos, Camino Chinquihue Km 6, Puerto Montt 5480000, Chile; edwin.niklitschek@ulagos.cl
*   Correspondence: fernando.garcia-solis@alumnos.ulagos.cl; Tel.: +34-690-884-935

**Abstract:** The two mammalian carnivores, puma (*Puma concolor*) and South American grey fox (*Lycalopex griseus*) were studied, in a remote area located in the humid temperate forest of the coastal range of southern Chile. A total of six locations were selected in three landscapes: pre-mountain range, mountain range, and coast. The chosen study locations are relevant because they correspond to threatened areas with different levels of human intervention., so they offer the ideal setting for studying how different species of carnivores respond to both human presence and activities. A dataset was collected for 24 months during 2016–2018 through photo-trapping (13 camera traps placed along 50 photo-trap stations). Wes estimated the apparent occurrence and relative abundance index (RAI) of the fauna registered, by means of generalized linear models to contrast those of an apex predator, such as the puma and a sympatric mesopredator, the South American grey fox, across the three landscapes. The ecological variables assessed were the RAI of the other carnivore considered, exotic carnivores such as dogs and cats, human intervention, farmland effect, prey availability, and habitat quality. The primary hypothesis was that the apparent occurrence and RAI of puma and fox would be positively associated with the RAI of prey and livestock and negatively with human intervention. On the other hand, the secondary hypothesis dealt with the interactions between puma and fox faced with different degrees of human intervention. The results showed that the apparent occurrence of the puma was statistically explained by location only, and it was highest at the mountain range. The apparent occurrence of foxes was explained by both puma apparent occurrence and relative integrated anthropization index (INRA), being highest in the pre-mountain range. Concerning the RAI of pumas, high values were yielded by location and fox RAI. For the RAI of foxes, they were location, puma RAI, and INRA. It can be suggested that eucalyptus plantations from the pre-mountain range could offer an adequate habitat for the puma and the fox, but not the coastal range, as the mountain range could be acting as a biological barrier. Due to the nature of the data, it was not possible to detect any relevant effect between the two carnivores' considered, between their respective preys, or the very abundant presence of dogs.

**Keywords:** camera-trapping; conservation puma; relative integrated anthropization index; INRA; South American grey fox

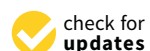



## 1. Introduction

Mammalian carnivores tend to have large home ranges, low densities, and slow growth rates, making them especially vulnerable to extinction [1–3]. Because of the lack of protection, habitat loss, and human action; most wild carnivores have undergone significant decreases in their abundance and diversity [1,2,4–8]. The conflict with humans is the leading cause of the decline in carnivore populations [9,10]. These conflicts happen mainly

because of suspected predation on livestock and on some wild species with trophy hunting interests [11]. These human carnivore conflicts are a worldwide problem [10,12] with plenty of examples of carnivores killing livestock or even attacking humans. Carnivores have an essential role in the community of which they are part of, primarily by regulating it through trophic cascades. Their effects can be produced by consumption or by behavior [13]. The consumption function is also called lethal and can directly regulate prey population size [8,14] and mesopredators (in the case of apex predators) [15,16]; or indirectly by providing carrion [17,18], promoting higher biodiversity levels [19], or even influencing soil composition [20]. Their effects by behavior can be direct and indirect as well: Directly by influencing prey behavior and habitat use [21,22], prey pack size [23], reproductive physiology [24], and natural selection [25]. Indirectly by modulating prey population dynamics [26,27], limiting herbivory, or maintaining plant diversity [28,29]. Therefore, carnivore protection is one of the priorities in biological conservation using the top-down approach [10].

Most ecological ecosystems are human-modified environments [4,30] due to urban development or exploitation of natural resources. Carnivores are affected by human activities in many different ways: by habitat fragmentation, physical barriers limiting gene flow, road death tolls, behavioral changes, dispersal, disease spreading, and exposure to poisons [3,31,32].

An essential aspect in carnivore conservation and management is based on their interactions in sympatry. It is crucial to understand the structure of the ecological community in which they are inserted [33], as it may influence the distribution, activity patterns, and or diet of the carnivores involved. The competitive exclusion principle proposes that two species with identical niches cannot coexist indefinitely; therefore, some degree of partitioning must materialize in the realized niche of coexisting species [33–36]. Such partitioning is commonly observed across time, space, and trophic axes. In addition, the particular association of coexistence established between apex predators and mesopredators should be considered. The latter being defined as those at intermediate trophic levels, where the former control the populations of the latter [15,16,33,37].

The present study is focused on the apparent occurrences, relative abundance indexes (RAI) [38], and connections of an apex predator, the puma (*Puma concolor*) and one mesopredator, the South American grey fox (*Lycalopex griseus*), considering ecological variables such as the abundance of others wild and exotic carnivores, human intervention, farmland effect, prey availability, and habitat quality. Consideration of dogs is also important as well because we observed abundant free-roaming individuals were observed, which may influence both native species (by predation, competition, disease transmission) [39–42], and livestock [43,44]. The chosen study locations are relevant, because they are threatened areas with different levels of human intervention. These features offer the ideal setting for studying of how different carnivore species respond to human presence and activities. The primary hypothesis was that the apparent occurrence and RAI of puma and fox would be positively associated with RAI of prey and livestock and negatively related to human intervention. On the other hand, the secondary hypotheses were related to the interactions between puma and fox faced with different degrees of human intervention.

In this work, the puma and fox apparent occurrence and RAI between three contrasting landscapes were compared, characterized by considerable differences in human population and intervention. Under this central hypothesis, lower RAI and apparent occurrence of both predators in the pre-mountain landscape are expected, which was more anthropized. Besides this, the authors were interested in assessing several secondary hypotheses that might explain the variability observed between localities. They included a negative relationship between puma and fox, positive effects of prey and livestock apparent abundance, and negative effects of humans and free-roaming dogs on the apparent occurrence and RAI of both carnivores. Nonetheless, the large collinearity between most of these explanatory variables and the small number of localities where they were tested precluded proper isolation of their effects, leading to shape the current assessment as an exploratory analysis.

## 2. Materials and Methods

The methodology chosen was camera-trapping, which is non-invasive, considers the wellbeing of the animals by minimizing the disturbance of their activities, and guarantee their safety. These methods have been increasingly used over the last 30 years [45], especially in the study of carnivores [46,47]. As shown, camera trapping is an effective method for wildlife surveys, is easy to use, allows to obtain information remotely, is resilient to weather conditions, and has a reasonable cost [45].

### 2.1. Study Site

The study area is located in the Valdivian Eco-region (40°–42° S) in the humid temperate forest of the coastal range [48]. Six locations from the Purranque Commune in Osorno Province of Los Lagos Region were selected across three landscapes: pre-mountain range (locations of Hueyusca and Los Riscos), mountain range (locations: slope and the peak), and coast (locations of San Pedro Bay and Manquemapu) (Figure 1). The latter two belong to the Lafken Mapu Lahual Indigenous Protected Area [49]. The Valdivian Rainforest is one of the top conservation priorities worldwide due to its high levels of endemism and biodiversity. The three landscapes studied have very different degrees of human intervention, which offers the chance to study and compare the effects of such intervention upon carnivore RAI and apparent occurrence. Due to a limited number of cameras available and logistic difficulties to access the study area, only two locations per landscape were chosen.

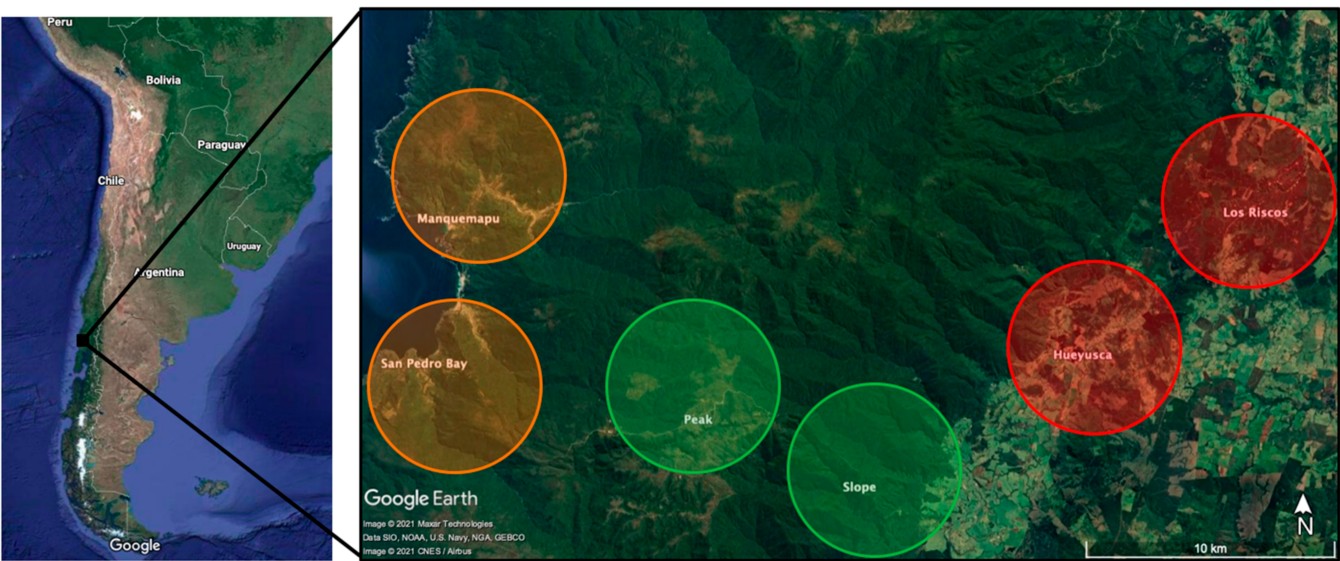

**Figure 1.** Study area (800 km² approximately). Coast locations: Manquemapu and San Pedro Bay (Orange). Mountain range locations: peak and slope (Green). Pre-mountain range locations: Hueyusca and Los Riscos (red).

The climate is rainy temperate, characterized by moderate temperatures (average of the coldest month is 7.5 °C, of the warmest month is 22 °C, with a yearly average over 10 °C [50,51]). Rains occur throughout the year, lacking a dry season [50,51]. During 2017, the rainiest month was August (289.4 mm) and the lowest precipitation was during November (22.8 mm), averaging 112 mm yearly [52].

The pre-mountain range is a human-dominated landscape, with small-family livestock owners and large patches of exotic plantations of eucalyptus (mainly Eucalyptus nitens and Eucalyptus globulus [53]) and pines (*Pinus* spp.). The location of Hueyusca has 399 inhabitants [54] whose main activity is related to small-scale livestock raising and agriculture. The location of Los Riscos has 130 inhabitants [54], mainly related to eucalyptus forestry practices. There are still fragments of deciduous forest of Patagonian oak (*Nothofagus obliqua* now *Lophozonia heterocarpa* [55]), and Chilean laurel (*Laurelia sempervirens*), coigüe

(*Nothofagus dombeyi*), and ulmo (*Eucryphia cordifolia*) mixed forest closer to the mountain range [56].

The mountain range is a more pristine landscape with a low human population and intervention. The location of slope has 89 inhabitants [54] and the peak has no official population records, five inhabited houses were observed nearby the sampling site. The vegetation is dominated by a mixed forest of coigüe with ulmo in the east slope, a narrow strip of Patagonian cypress (*Fitzroya cupressoides*) at the top, and tineo (*Weinmannia trichosperma*) with tepa (*Laureliopsis philippiana*) on the west slope [56].

The coast landscape has a few small indigenous settlements, whose main activity is fishing, complemented by the collection and handwork of local wood [57]. The vegetation surrounding these settlements is dominated by tineo and tepa [56]. No official population records are available for San Pedro Bay or Manquemapu, though the local government estimates they have about 40 and 100 inhabitants, respectively.

An essential feature of our study area is their inhabitants, as there are several native communities of Huilliche natives (people from the south) they are one of the several Mapuche ethnic groups, whose lives are linked to nature and its resources, especially the Patagonian cypress. Wood handicraft is one of their main activities, but they also work the land, raise livestock and crops, or do fishing if they live close to the coast, all in a traditional fashion [57,58].

### 2.2. Study Design

A total of 13 camera traps (Bushnell 8MP Trophy Cam HD Hybrid Trail Camera with Night Vision) were individually placed along 50 photo-trap stations to maximize the number of cameras: 14 in the coast (6 in Manquemapu and 8 in San Pedro Bay), 16 in the mountain range (8 in the slope and 8 at the peak), and 20 in the pre-mountain range (9 in Hueyusca and 11 in Los Riscos). The photo-trap period lasted from April 2016 to March 2018, with a survey period of 5772 camera days. The cameras were placed between 50–70 cm high [59–62] along secondary paths, randomly within the specific location [59,63] and with a minimum separation distance of 3 km among them [46,59,61,64], to promote the spatial independence in detections. To further optimize the use of cameras, specific attractants for carnivores were applied [62,65–67], chiefly commercial Eurasian lynx (*Lynx lynx*) urine.

The camera traps were placed on large-diameter trees to prevent or hinder their removal by humans, which was a problem during our study. Another measure to prevent these events was the use of two locks and one chain per camera. Once a month the status of the cameras, battery levels, and memory cards were checked, and their contents were transferred if it necessary for the research design. The camera settings corresponded to the following: mode: camera, image size: 5M pixel, image format: full screen, capture number: 3 photo, led control: medium, camera name: input, interval: 5 s, sensor level: low, NV shutter: low, camera mode: 24 h, format: execute, time stamp: on, and field scan: off. Those photographic records with animals were considered as independent events when images contained species within a 60-minute period. If another animal of the same species was captured in this time window, it was not registered unless it could be recognized as a different individual [60,64,68].

To identify which of the three Chilean fox species known were recorded, the photos were reviewed by the study team, determining the species positively as South American grey fox (*Lycalopex griseus*), which agreed with the bibliography labelling it as more of a lowland animal than culpeo fox (*Lycalopex culpaeus*), the latter being more of a mountain dweller of the Andes range [69,70].

### 2.3. Data and Statistical Analysis

The data were analyzed with the statistical program R [71]. RAI by species, location, season, and camera was computed as the total number of independent and recognizable pictures of each species recorded by a single camera placed at a particular location within a single season. As the number of deployment days was variable between cameras, locations,

and seasons, RAI was standardized to a fixed 100-day period [38,72]. The apparent occurrence was computed as a dichotomic variable indicating presence for all RAI values > 0 and absence otherwise. Apparent occurrence and RAI of puma and fox were then analyzed using a generalized linear models (GLMs) framework [73] (Table 1). On the other hand, a binomial distribution was inherent for apparent occurrence data, a zero-inflated negative binomial distribution [74] was used to analyze RAI responses, respectively. Model assumptions were assessed using a simulated residuals approach [75] as implemented in the R package DHARMa [76]. Dichotomic uses of *p*-values were purposely avoided following recommendations made by the American Statistical Association [77] and a growing number of scientists worldwide [78].

**Table 1.** Model used to estimate the apparent occurrence and RAI of puma and fox. AO = Apparent occurrence.

| Puma | Model in R |
|---|---|
| Puma AO | gpuma.bin = glmer(formula = Puma.bin~Habitat + Dog + Fox + PumaPrey + Livestock + HumanPresence + Inra + (1 | Locality), family = "binomial", data = data3, na.action = "na.pass") |
| Fox AO | gzorro.bin2 = glmer(formula = Zorro.bin~Habitat + Dog + Puma + FoxPrey + Livestock + HumanPresence + Inra + (1 | Locality), family = "binomial", data = data3, na.action = "na.pass") |
| Puma RAI | gpuma.ab = glmmadmb(Puma~ Habitat + Dog + Fox + PumaPrey + Livestock + HumanPresence + Inra + (1 | Locality), data = data3, zeroInflation = TRUE, family = "nbinom") |
| For RAI | gzorro.ab = glmmadmb(Zorro~Habitat + Dog + Puma + FoxPrey + Livestock + HumanPresence + Inra + (1 | Locality), data = data3, zeroInflation = TRUE, family = "nbinom") |

In accordance with the primary hypothesis, landscape effects upon RAI and apparent occurrence of puma and fox were assessed by means of marginal likelihood ratio-tests [79]. To properly isolate landscape effects, GLMs used for this purpose also included season and location effects. This sampling design-based analysis was followed by an exploratory analysis of the secondary hypotheses, where locality and seasonal effects were replaced by six quantitative variables: dog (*Canis familiaris*), competitor, prey and livestock apparent relative abundances (records/100 camera-days), human presence and degree of anthropization. Competitor RAI corresponded to either puma or fox standardized records, while apparent prey abundances summed over European hare (*Lepus europaeus*) and pudu (*Pudu puda*) records for both predators, plus red deer (*Cervus elaphus*) records for puma. Livestock apparent abundance summed over apparent abundances of horse (*Equus caballus*), cattle (*Bos taurus*), sheep (*Ovis aries*) and pig (*Sus scrofa domestica*). Human presence was indexed summing over people, vehicles, and machinery records.

Anthropization was indexed through the relative integrated anthropization index (INRA) [80,81], computed after assigning intervention values (0.000–1.000) reflecting the use or land cover by subunits of analysis (SUA) [80,82], 0 being no-intervention level and 1 maximum intervention level. In this case, $0.1 \times 0.1$ km quadrants were used (aerial images from Google Earth). The categories assigned for land use or land cover were the following: native vegetation (0.000), native vegetation + clearing (0.125), native vegetation + clearing + crops (0.250), native vegetation + crops (0.375), clearing (0.5000), clearing + crops (0.625), cultivation (0.750), rural population (0.850), and urban nucleus (1.000). Once the SUA values were obtained, the INRA value of the analysis units (UA) was obtained as:

$$INRA = \left( \sum SUA' / n \right) \cdot 100 \qquad (1)$$

where $\sum SUA'$ = the sum of the partial anthropization value of all SUA and n = total number of SUAs.

The exploratory analysis was performed following a two-steps approach. First, a deviance analysis followed by marginal likelihood ratio-tests was used to assess all main

effects of habitat and the six quantitative variables considered at once [79]. Second, a multi-model comparison approach based on second-order Akaike's information criterion [82] was used to compare and rank all possible combinations of habitat and quantitative variables using the second-order Akaike's information criterion and Akaike's weight, which was interpreted as the probability of being the most parsimonious model within the set of candidate models being compared [82].

## 3. Results

A total of 3611 records (Tables A1 and A2) were obtained with an average of 55 per camera, location, and year. Because of the low number, data from the coastal range were removed from further analyses. Among carnivores, it was possible to detect: kodkod (*Leopardus guigna*), Molina's hog-nosed skunk (*Conepatus chinga*), puma (*Puma concolor*), South American grey fox (*Lycalopex griseus*), and two exotic species, dog (*Canis familiaris*), and cat (*Felis catus*).

The relative integrated anthropization Index (INRA) showed the general trend of highest values appearing in the pre-mountain range, followed by the coast, and ended by the mountain range (Table 2).

**Table 2.** Relative integrated anthropization index (INRA) with respective SUAs (subunits of analysis) by location in the coastal range of southern Chile. (C: coast, MR: mountain range, PMR: pre-mountain range).

| Units of Analysis | Landscape | SUA1 | SUA2 | SUA3 | SUA4 | SUA5 | SUA6 | SUA7 | SUA8 | SUA9 | INRA |
|---|---|---|---|---|---|---|---|---|---|---|---|
| Manquemapu | C | 0.125 | 0 | 0.125 | 0 | 0 | 0.125 | 0 | 0 | 0 | 4.167 |
| San Pedro | C | 0 | 0.5 | 0 | 0 | 0 | 0.5 | 0 | 0.125 | 0 | 12.500 |
| Peak | MR | 0 | 0.125 | 0 | 0 | 0.5 | 0.125 | 0 | 0 | 0 | 8.333 |
| Slope | MR | 0 | 0 | 0 | 0 | 0 | 0 | 0 | 0 | 0.125 | 1.389 |
| Hueyusca | PMR | 0.25 | 0.125 | 0.75 | 0 | 0.25 | 0.625 | 0.125 | 0.375 | 0.125 | 29.167 |
| Los Riscos | PMR | 0.125 | 0.125 | 0.125 | 0.25 | 0.375 | 0.75 | 0.375 | 0.25 | 0.75 | 34.722 |

### 3.1. Exploratory Analysis of Primary Hypothesis

Significant differences in RAI and apparent puma occurrence were observed between landscapes and locations (Figures 2 and 3). Although evident landscape effects (Table 3) resulted from the complete absence of puma records in the coast (Figure 2), no apparent differences were found when locality means were compared between the pre-mountain and mountain landscapes. Thus, the highest puma apparent occurrence and RAI means were not consistently observed in the mountain landscape, although their maximum values did occur in the less anthropized location of the slope (Figures 2 and 3, Table 2).

**Table 3.** Marginal likelihood ratio tests for the effects of landscape, season, landscape: location, and landscape: Season on the apparent occurrence and RAI of puma (*Puma concolor*) and South American grey fox (*Lycalopex griseus*) in six locations of the Southern Chile coastal range. *p*-values < 0.1 highlighted in bold.

| | | Puma | | | | Fox | | | |
|---|---|---|---|---|---|---|---|---|---|
| | Degrees of Freedom | Apparent Occurrence | | RAI | | Apparent Occurrence | | RAI | |
| | | LR $X^2$ | $p$ (>$X^2$) | LR $X^2$ | $p$ (>$X^2$) | LR $X^2$ | $p$ (>$X^2$) | LR $X^2$ | $p$ (>$X^2$) |
| Landscape | 2 | 19.08 | **<0.001** | 24.51 | **<0.001** | 8.32 | **0.016** | 12.10 | **0.002** |
| Season | 3 | 2.58 | 0.461 | 4.14 | 0.247 | 3.74 | 0.291 | 3.68 | 0.298 |
| Landscape: Location | 3 | 1.64 | 0.651 | 6.90 | **0.075** | 10.33 | **0.016** | 5.79 | 0.123 |
| Landscape: Season | 6 | 3.36 | 0.762 | 6.36 | 0.384 | 6.92 | 0.329 | 8.70 | 0.191 |

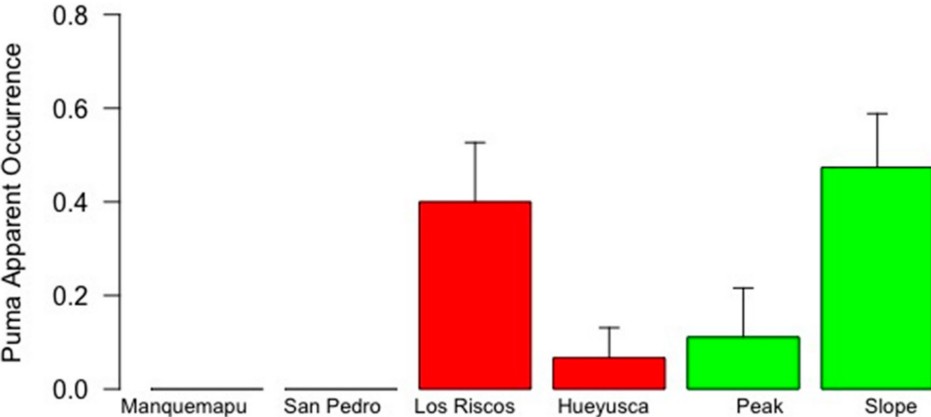

**Figure 2.** Puma apparent occurrence in six locations of the coastal range of southern Chile (whiskers represent 1 standard error). White, red and green bars identify coast, pre-mountain range and mountain range landscapes, characterized by intermediate, high and low degrees of human intervention, respectively.

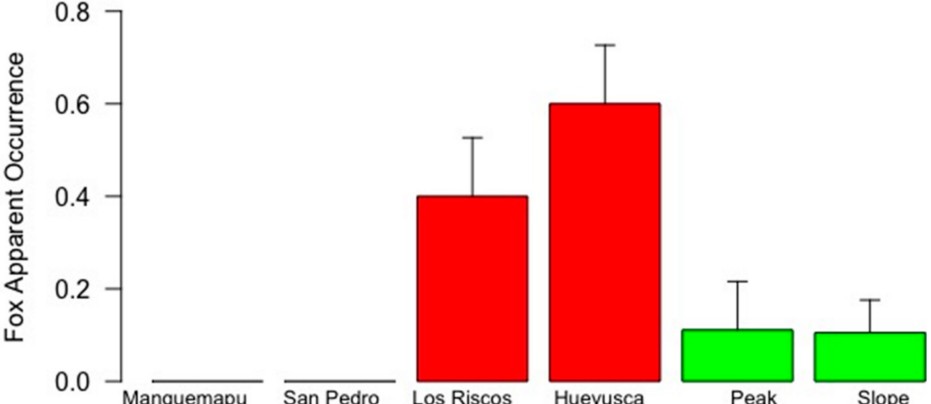

**Figure 3.** Fox apparent occurrence in six locations of the coastal range of southern Chile (whiskers represent 1 standard error). White, red and green bars identify coast, pre-mountain range and mountain range landscapes, characterized by intermediate, high and low degrees of human intervention, respectively.

Variability in fox apparent occurrence appeared more clearly linked to landscape features (Table 2, Figure 4), with higher mean values found in the pre-mountain range and no evidence of fox presence in the coast. Variability in fox RAI was inconsistent between landscapes, with maximum values in Hueyusca (pre-mountain) followed by the peak (mountain). Thus, as before, maximum RAI and apparent occurrence values were not found in the less disturbed mountain range landscape, nor the less anthropized locations of the slope and Manquemapu (Figures 4 and 5, Table 2).

### 3.2. Exploratory Analysis of Secondary Hypothesis

Deviance analysis of secondary hypotheses showed that none of the variables being considered exhibited relevant marginal effects explaining variability in puma apparent occurrence ($p \geq 0.403$, Table 4), whereas some evidence was found ($p = 0.074$, Table 4) of positive effects from the RAI of fox on that of puma. Model selection procedures also failed to identify an informative model for explaining apparent puma occurrence (Table 5). The most informative model which included positive effects from fox RAI as its only explanatory variable has a probability (AICc-w) of just 0.07 (Table 5) and explained only 8% of the deviance. The other four alternative models, including the null model, all received weak support from the data ($\Delta$AICc $\leq 2$, Table 5). Model selection results for puma RAI were more conclusive. The most comprehensive model one that considered the negative effects of INRA and positive effects of livestock apparent abundance exhibited a probability of 0.54 and explained 37% of the observed deviance (Table 6).

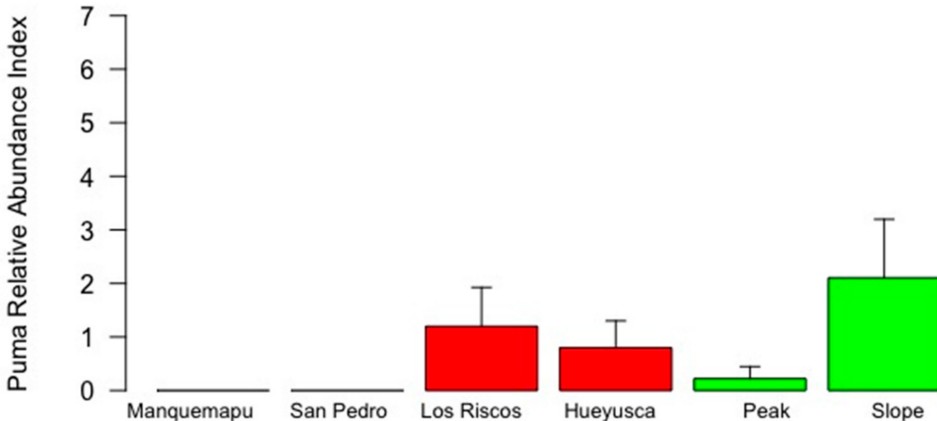

**Figure 4.** Puma RAI in six locations of the coastal range of southern Chile (whiskers represent 1 standard error). White, red and green bars identify coast, pre-mountain range and mountain range landscapes, characterized by intermediate, high and low degrees of human intervention, respectively.

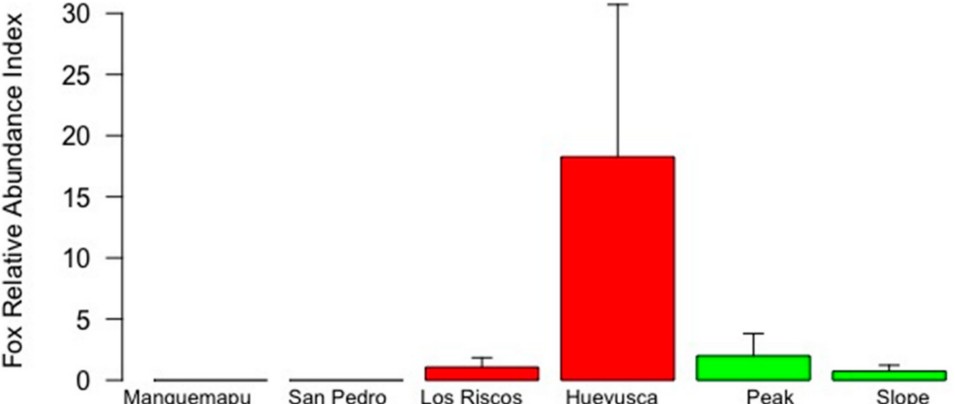

**Figure 5.** Fox RAI in six locations of the coastal range of southern Chile (whiskers represent 1 standard error). White, red and green bars identify coast, pre-mountain range and mountain range landscapes, characterized by intermediate, high and low degrees of human intervention, respectively.

**Table 4.** Marginal likelihood ratio tests for secondary hypotheses, which considered the effects of the six quantitative variables considered on the apparent occurrence and RAI of puma (*Puma concolor*) and South American grey fox (*Lycalopex griseus*) in six locations of the Southern Chile coastal range. *p*-values < 0.1 highlighted in bold.

| | Degrees of Freedom | Puma | | | | Fox | | | |
|---|---|---|---|---|---|---|---|---|---|
| | | Apparent Occurrence | | RAI | | Apparent Occurrence | | RAI | |
| | | LR $X^2$ | $p$ (>$X^2$) | LR $X^2$ | $p$ (>$X^2$) | LR $X^2$ | $p$ (>$X^2$) | LR $X^2$ | $p$ (>$X^2$) |
| Dog | 1 | 0.388 | 0.533 | 0.223 | 0.637 | 0.03 | 0.864 | −1.126 | 1.000 |
| Competitor | 1 | 2.793 | **0.095** | 5.832 | 0.016 | 4.527 | **0.033** | 17.928 | **<0.001** |
| Prey | 1 | 0.602 | 0.438 | 0.257 | 0.612 | 0.974 | 0.324 | 0.134 | 0.714 |
| Livestock | 1 | 0.701 | 0.403 | 0.233 | 0.629 | 2.941 | **0.086** | 0.298 | 0.585 |
| Human presence Index | 1 | 0.007 | 0.934 | 0.114 | 0.735 | 0.006 | 0.939 | 0.768 | 0.381 |
| INRA | 1 | 0.136 | 0.712 | 0.973 | 0.324 | 3.047 | **0.081** | 10.112 | **0.001** |

**Table 5.** Exploratory analysis of potential explanatory variables for observed variability in the apparent occurrence of puma (*Puma concolor*). Only the best five models, as ranked by AICc values, are shown. Positive/negative signs indicate variables and effects considered by each model. H = landscape, L = location, LST = livestock, HPI = human presence index, INRA = relative integrated anthropization index, D2 = explained deviance, K = number of estimated parameters, logLik = log$_e$ likelihood, AICc = second order Akaike information criterion, AICc-W = Akaike weight.

| Model | Dog | Prey | Fox | LST | HPI | INRA | D$^2$ | K | logLik | AICc | ΔAICc | AICc-W |
|-------|-----|------|-----|-----|-----|------|-------|---|--------|------|-------|--------|
| 1 | | | +1.50 | | | | 8.23% | 3 | −32.59 | 71.63 | 0.00 | 0.07 |
| 2 | | | | +1.27 | | | 8.09% | 3 | −32.64 | 71.72 | 0.09 | 0.07 |
| 3 | | | | | | | 3.82% | 2 | −33.95 | 72.13 | 0.50 | 0.05 |
| 4 | +095 | | | | | | 5.89% | 3 | −33.32 | 73.09 | 1.46 | 0.03 |
| 5 | | | | +0.97 | +0.77 | | 9.15% | 4 | −32.30 | 73.36 | 1.73 | 0.03 |

**Table 6.** Exploratory analysis of potential explanatory variables for observed variability in in the RAI of puma (*Puma concolor*). Only the best five models, as ranked by AICc values, are shown. Positive/negative signs indicate variables and effects considered by each model. Codes are the same as in Table 5.

| Model | Dog | Prey | LST | HPI | INRA | D$^2$ | K | logLik | AICc | ΔAICc | AICc-W |
|-------|-----|------|-----|-----|------|-------|---|--------|------|-------|--------|
| 1 | | | +0.17 | | −0.17 | 36.58% | 6 | −65.39 | 144.42 | 0.00 | 0.54 |
| 2 | | | +0.17 | | −0.29 | 37.04% | 7 | −64.86 | 145.97 | 1.55 | 0.25 |
| 3 | | | +0.15 | | | 32.71% | 6 | −67.14 | 147.92 | 3.50 | 0.09 |
| 4 | | +0.24 | +0.24 | −0.23 | −0.17 | 45.23% | 8 | −64.70 | 148.34 | 3.91 | 0.08 |
| 5 | +0.21 | | +0.16 | | −0.26 | 37.36% | 7 | −66.78 | 149.81 | 5.38 | 0.04 |

Deviance analysis applied to fox records showed more significant marginal effects of livestock apparent abundance and INRA on apparent fox occurrence, and puma RAI and INRA on fox RAI. As found for puma, model selection failed to identify a distinct set of explanatory variables accounting for variability in fox apparent occurrence. Five models received similar support from the data (ΔAICc ≤ 2): the most informative exhibited probabilities between 0.17 and 0.23 and explained 34–39% of the observed deviance (Table 7). All these models included positive effects of INRA and livestock apparent abundance. Some of them also included positive effects of prey, dog and puma RAI (Table 7). For the RAI of fox, three potentially informative models were identified, with probabilities between 0.22 and 0.33 and explaining 43–46% of the observed deviance (Table 8). While these models included positive effects of INRA and livestock apparent abundance, the top one also included negative effects of apparent dog abundance (Table 8).

**Table 7.** Exploratory analysis of potential explanatory variables for observed variability in the apparent occurrence of South American grey fox (*Lycalopex griseus*). Only the best five models, as ranked by AICc values, are shown. Positive/negative signs indicate variables and effects considered by each model. Codes are the same as in Table 5.

| Model | Dog | Prey | Puma | LST | HPI | INRA | D$^2$ | K | logLik | AICc | ΔAICc | AICc-W |
|-------|-----|------|------|-----|-----|------|-------|---|--------|------|-------|--------|
| 1 | | + 2.44 | | + 3.18 | | +2.66 | 36.72% | 5 | −22.65 | 56.46 | 0.00 | 0.23 |
| 2 | | +2.36 | +1.35 | +2.81 | | +2.98 | 39.24% | 6 | −21.48 | 56.60 | 0.14 | 0.21 |
| 3 | +2.23 | | +1.53 | +2.41 | | +3.07 | 39.06% | 6 | −21.56 | 56.77 | 0.31 | 0.20 |
| 4 | | | | +3.13 | | +2.92 | 33.61% | 4 | −24.04 | 54.84 | 0.38 | 0.19 |
| 5 | | | +1.29 | +2.81 | | +3.22 | 36.09% | 5 | −22.94 | 57.03 | 0.57 | 0.17 |

**Table 8.** Exploratory analysis of potential explanatory variables for observed variability in the RAI of South American grey fox (*Lycalopex griseus*). Only the best five models, as ranked by AICc values, are shown. Positive/negative signs indicate variables and effects considered by each model. Codes are the same as in Table 5.

| Model | Dog | Prey | LST | HPI | INRA | D2 | K | logLik | AICc | ΔAICc | AICc-W |
|---|---|---|---|---|---|---|---|---|---|---|---|
| 1 | −0.03 | +0.05 | +0.05 | | +0.05 | 46.01% | 8 | −79.93 | 178.79 | 0.00 | 0.33 |
| 2 | | +0.03 | +0.03 | | +0.05 | 42.95% | 7 | −81.52 | 179.29 | 0.50 | 0.26 |
| 3 | | +0.03 | +0.05 | −0.02 | +0.06 | 45.23% | 8 | −80.34 | 179.62 | 0.83 | 0.22 |
| 4 | −0.03 | +0.05 | +0.06 | −0.01 | +0.06 | 46.44% | 9 | −79.70 | 181.14 | 2.35 | 0.10 |
| 5 | −0.03 | +0.05 | +0.05 | | +0.05 | 46.09% | 9 | −79.88 | 181.52 | 2.72 | 0.09 |

## 4. Discussion

The absence of puma and fox records obtained in the two locations from the coast landscape (Manquemapu and San Pedro Bay) was surprising. Nonetheless, it matched results from a parallel study conducted by us, which showed that carnivore feces were scarce in these locations. These sites are isolated, weakly intervened, and with small settlements of fishermen and wood handcrafters (INRA values of 4.167 and 12.500, respectively). When the study was designed, it was assumed that the mountain range would act as a biological corridor [83], but the current situation probably is the opposite, acting as a barrier and limiting dispersal from the coast landscape. In the past, the entire mountain range suffered from several big fires [57]; some people think they came about by natural causes and others that they were man-made to acquire the burned wood from Patagonian cypress, which is protected as a natural monument and can only be exploited when burnt (independently of cause). Currently, the Patagonian cypress forest at the peak is quite open, full of dead trees, a few survivors, and some recruits (F. García-Solís, personal observation). Unfortunately, Patagonian cypress trees take longer to grow, living up to 3600 years [84]. All this renders the peak location of the mountain range a harsh environment, with almost no shelter for herbivores, thus limiting carnivore presence.

Camera trapping of unmarked species can be challenging, as it is difficult to use capture-recapture methods when assessing their relative abundances and could have biased inference estimating abundances [85,86]. In our study, the two carnivore species were unmarked, thus we assumed equal detectability and potential bias, as camera traps cannot record all animal presences in an area [87]. Their camera records were considered as independent events when consecutive images that contained the same species were recognizable as different individuals, a method used in several studies [60,64,68]. The use of lures is a widespread method in camera trapping, but optimizing the detectability of a target species can produce bias in calculating abundances, as the species behavior may be altered, or some species may be attracted whereas others may be repelled [88–91].

### 4.1. Relative Integrated Anthropization Index (INRA)

The working hypothesis about habitat quality was related to the fact that the locations from the pre-mountain range would have the highest INRA levels, followed by coast and then by mountain range. Our results supported this mostly, except for the Manquemapu and peak locations. The former had lower INRA, affording better habitat quality than the latter. This lower INRA may be accounted for by the operation of the Manquemapu Management Plan, regulated by its Mapuche Huilliche community. This plan considers zoning areas of human use, dead Patagonian cypress recovery harvest, sustainable management, and collection of marine resources [92]. Although the peak has low human intervention, its forest may offer lower habitat quality owing to its past fire history. The current landscape is a very open forest, full of burned trunks which some of which show small brunches with leaves, this harsh environment could be a barrier to the dispersal of carnivores.

### 4.2. Predator Apparent Occurrence

Since the use of occupancy or co-occurrence models, was not supported by the data, the use of the apparent occurrence was selected. Los Riscos (pre-mountain range) is characterized by the presence of exotic plantations of eucalyptus (*Eucalyptus nitens* and *Eucalytus globulus*), which they are not native forests still afford a habitat for the puma [53], providing shelter from humans in the surroundings, and probably also food, by being populated by hare and pudu. Further, in that particular landscape, there is a vital remnant of native forest [53,93,94], which provides habitat for the puma's prey. In addition, there is the presence of livestock, which pumas may perceive as a potential food resource. The slope location from the mountain range landscape is characterized by scarce human presence and low activity and preserves most of its native vegetation, rendering it relatively unaltered by humans, which may explain the high puma apparent occurrence. Our results from the marginal likelihood ratio test showed that there was not a significant effect of human activity on puma, pumas may tolerate human presence better than expected. In addition, the effect from prey could not show to be influential either, this could be explained for the prey cannot be detected by the cameras since carnivorous attractant was used. In parallel, the models from the exploratory analysis showed some positive relations for RAI of fox and livestock, both being a potential food resource for puma.

Apparent fox occurrence was influenced by RAI of puma, livestock and INRA. These values were higher in the pre-mountain than in the mountain range. The mesopredator release hypothesis [16] may explain the higher fox presence in the former landscape because the higher human activity may interfere with apparent puma occurrence. A complementary explanation is that foxes, being mesopredators with smaller size may tolerate environments with higher human activity [95]. This parallels the positive effect of INRA on fox apparent occurrence. Alternatively, the presence of puma can facilitate the presence of foxes since the puma behavior of burying its prey after eating to store it for later; this buried prey being subsequently scavenged by foxes [96–98], explaining the positive correlation between them. The exploratory analysis supported the positive effects of INRA and puma on fox. Additionally, South American grey foxes are known to visit exotic plantations due to potential prey such as rodents and hares [99].

### 4.3. Predator Relative Abundance Indexes

Relative Abundance indices are not necessarily the most informative about abundance species and can have some weaknesses such as: be biased due to the different detection among species, especially in elusive ones; species with extensive home range are more detected, increasing RAI values; and bias due to the different responses to the camera setup among the species [100]. Puma RAI was influenced by the variables of location and fox RAI. Los Riscos and slope present higher values, probably due to low levels of human activity and restricted pass policy, in addition to the presence of livestock as potential prey (our data showed a positive but not so strong correlation. The positive and relevant correlation between RAI of fox and puma could be explained by intraguild predation, which is an extreme form of interspecific competition when species that act as competitors also function as predators [101–103]. In this case, the puma is a potential predator of foxes, the latter's RAI may improve an increase in that of puma. The exploratory analysis showed a negative relation with INRA, and one model showed a negative relation also with human activity, which could be explained by the sensitivity of puma to habitat quality.

Despite large differences in fox RAI between landscapes and locations, our analysis suggests that puma RAI and INRA were positively associated with fox RAI. For instance, Hueyusca showed both the highest fox RAI and high levels of human presence and activity (INRA = 29.167), suggesting once more that foxes may flourish in such anthropized situation. The exploratory analysis supported the survivorship of foxes in human-intervened environments and showed a positive relation of foxes with their prey and the presence of livestock.

It was noteworthy that, even though the cameras traps registered numerous dogs, they were not identified as important ecological variables in any of the models depicting RAI and the apparent occurrence of puma and fox. This result was unexpected, as the impact of free-roaming dogs over wildlife by predation, activity alteration (fear-related), hybridization, and spreading of diseases is well known [42,104]. The present data show that dog numbers were larger in the pre-mountain range whereas those of puma were so in the mountain range. Thus, these two carnivores were segregated over the spatial axis so that dogs may not have an important effect over pumas. Nevertheless, dogs and foxes are abundant in the pre-mountain range, but even if they share space, they are segregated over time, foxes being more active during the night and dogs during the day [8,104].

It can be suggested that eucalyptus plantations in the pre-mountain range could offer an adequate habitat for the puma and the fox due to the presence of shelter from humans from the surroundings and prey availability such as hare, pudu, rodents, and potentially livestock. This was not the case of the coastal range, where we obtained almost no animal records, so it is possible that the mountain range could be acting as a biological barrier rather than a biological corridor. Due to the nature of the present data, it was no possible to detect any relevant effect between the two coexisting carnivores, between their respective prey, or the very abundant presence of dogs. Consequently, we recommend further studies in this specific area and habitats, improving the sampling efforts by implementing a considerable number of cameras and for more extended periods to obtain better data and clearer the relations and conclusions.

**Author Contributions:** Conceptualization, F.G.-S., J.R.R. and E.J.N.; methodology, F.G.-S., J.R.R. and E.J.N.; software, F.G.-S. and E.J.N.; validation, J.R.R. and E.J.N.; formal analysis, E.J.N.; investigation, F.G.-S., J.R.R. and E.J.N.; resources, J.R.R. and E.J.N.; data curation, E.J.N.; writing -original draft preparation, F.G.-S.; writing -review and editing, F.G.-S., J.R.R. and E.J.N.; visualization, J.R.R. and E.J.N.; supervision, J.R.R.; project administration, J.R.R.; funding acquisition, J.R.R. All authors have read and agreed to the published version of the manuscript.

**Funding:** This research was funded by ANID PIA/BASAL FB0002.

**Data Availability Statement:** Not applicable.

**Acknowledgments:** This research was supported by a doctoral fellowship from Universidad de Los Lagos (Chile) to Fernando García-Solís Marchant. The equipment used was provided by LABECOL (Ecology laboratory) from the Department of Biological Sciences and Bio-diversity of Los Lagos University and CAPES (Center of Applied Ecology and Sus-tainability. Special mention goes to the Fabián M. Jaksic who helped in the reviews of this manuscript; and the Municipality of Purranque, specifically to its Environmental Department chief, Carlos Oyarzún, who provided help in the logistic aspects of the study while funding most of the transportation. Finally, thanks to Anchile Lda. (Robinson Hapette), which allowed the entrance and camera installation in their plantation lands.

**Conflicts of Interest:** The authors declare no conflict of interest.

## Appendix A

**Table A1.** Records of animal species by landscape in the coastal range of southern Chile.

| Common Name/Category | Species | Pre-Mountain Range | Mountain Range | Coast | Total |
|---|---|---|---|---|---|
| **Mammalia, Order Carnivora** | | | | | |
| Dog | *Canis familiaris* | 753 | 70 | 6 | 829 |
| Domestic cat | *Felis catus* | 1 | 1 | 0 | 2 |
| Kodkod | *Leopardus guigna* | 9 | 1 | 0 | 10 |
| Molina's Hog-nosed skunk | *Conepatus chinga* | 0 | 4 | 0 | 4 |
| Puma | *Puma concolor* | 18 | 26 | 0 | 44 |
| South American grey fox | *Lycalopex griseus* | 157 | 12 | 0 | 169 |
| **Mammalia, Order Cetartiodactyla** | | | | | |

**Table A1.** *Cont.*

| Common Name/Category | Species | Pre-Mountain Range | Mountain Range | Coast | Total |
|---|---|---|---|---|---|
| Southern pudu | *Pudu puda* | 1 | 4 | 0 | 5 |
| Red deer | *Cervus elaphus* | 4 | 1 | 0 | 5 |
| **Mammalia, Order Lagomorpha** | | | | | |
| European hare | *Lepus europaeus* | 97 | 5 | 0 | 102 |
| **Mammalia, Order Rodentia** | | | | | |
| Unidentified rodent | Muridae? | 0 | 1 | 0 | 1 |
| **Aves, Order Pelecaniformes** | | | | | |
| Buff-necked ibis | *Theristicus caudatus* | 1 | 0 | 0 | 1 |
| **Aves. Order Cathartiformes** | | | | | |
| Black vulture | *Coragyps atratus* | 0 | 2 | 0 | 2 |
| **Aves, Order Strigiformes** | | | | | |
| Owl | Unknown Strigidae | 1 | 0 | 1 | 2 |
| **Aves, Order Falconiformes** | | | | | |
| Southern caracara | *Caracara plancus* | 9 | 2 | 0 | 11 |
| **Aves, Order Passeriformes** | | | | | |
| Austral thrush | *Turdus falcklandii* | 5 | 16 | 0 | 21 |
| **Aves, Order Columbiformes** | | | | | |
| Chilean pigeon | *Patagioenas araucana* | 10 | 1 | 0 | 11 |
| **Aves, Order Apodiformes** | | | | | |
| Unidentified hummingbirds | Unknown Trochilidae | 0 | 5 | 0 | 5 |
| **Livestock** | | | | | |
| Cow | *Bos taurus* | 425 | 217 | 42 | 684 |
| Calf | *Bos taurus* | 118 | 48 | 9 | 175 |
| Domestic horse | *Equus caballus* | 67 | 6 | 84 | 157 |
| Domestic foal | *Equus caballus* | 1 | 0 | 0 | 1 |
| Domestic pig | *Sus scrofa domestica* | 42 | 1 | 0 | 43 |
| Domestic sheep | *Ovis aries* | 16 | 0 | 0 | 16 |
| **Indeterminate animals** | | 33 | 2 | 0 | 35 |

**Table A2.** Records of human presence by landscape in the coastal range of southern Chile.

| Human Presence Sign | Species | Pre-Mountain Range | Mountain Range | Coast | Total |
|---|---|---|---|---|---|
| Machinery | | 18 | 0 | 0 | 18 |
| People | *Homo sapiens* | 435 | 143 | 244 | 822 |
| Vehicle | | 544 | 47 | 0 | 591 |

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
