# Peer review of "Occurrence and Abundance of an Apex Predator and a Sympatric Mesopredator in Rural Areas of the Coastal Range of Southern Chile"

_land, doi:10.3390/land11010040_

Round 1
Reviewer 1 Report
The authors have given a convincing answer to the questions raised in my review, and made the changes requested satisfactorily.
Reviewer 2 Report
The revised manuscript is not particularly novel, but it is scientifically sound.
Author Response
Please see the attachment.

This manuscript is a resubmission of an earlier submission. The following is a list of the peer review reports and author responses from that submission.
Round 1
Reviewer 1 Report
Overall, this manuscript is scientifically sound but there is room for improvement. My main suggestions are:
- Have a native English speaker review the paper for grammatical errors and misspelled words. There are numerous instances of these (for example, "Wes" in line 18 and "can be directly regulate" in line 56, etc).
- The rationale behind this manuscript would be much stronger if the authors presented a hypothesis driven approach. Instead of running the exploratory analyses presented, it would be great to see your initial hypotheses tested. For example, when you started the study, what were you trying to answer? Did you think fox abundance would be positively or negatively correlated with puma abundance? Did you think pumas would be correlated with livestock abundance? You should state your initial assumptions in the introduction and then test them specifically in your models. The presentation of the manuscript as is is very exploratory and the introduction is very general, which gives the reader the impression that the data were not collected to answer a specific question/questions.
- I disagree with the parsimonious model approach for these data. If there are variables that you think are influential in the distribution of these carnivore species, they should be assessed whether or not they make the models more parsimonious. For example, prey abundance is a natural predictor variable for carnivores. Yet, that variable was removed from GLMs due to AIC values, so we do not see the effect of that variable on carnivore abundance/occurrence. It seems like a lost opportunity to evaluate a variable of interest.
- Any variable that is discussed at length in the text should be tested in the models. For example, extensive discussion of conflict with livestock (line 65) is irrelevant since the authors did not evaluate livestock presence in GLMs.
- The map can be improved.
- Inset map should clearly show where the zoomed in area is.
- Add labels directly to camera sites instead of writing them out in the figure caption.
- Color code sites by location.
- The authors make broad conclusions with their data that are often stretching.
- For example, this study did not assess assemblages (line 105); it assessed the presence of 2 species.
- Text states that the mountains act as a barrier to movement of individuals (line 432), but these data are not fine enough (not enough cameras, did not detect individuals, etc) to make such a statement about behavior of individuals or their movement.
- Both the abstract and the main text call for further research on wildlife in human dominated landscapes. This is an unnecessary suggestion as there is extensive ongoing research in this field. However, such research is understudied in Latin America, so a call for more research in local areas is appropriate.
Reviewer 2 Report
General: The authors used camera traps as the main method for field data collection, as these tools are being increasingly used in field ecology. Their application, though, should be carefully planned and designed based on the hypotheses to be tested and the models to apply to analyze the data collected. The authors fail to explicitly introduce the hypotheses they want to test, and subsequently the description of the study design is poorly described.
Introduction: Not clear why the author try to relate the presence and abundance of two sympatric predators to broad issues such ad human-wildlife conflicts and habitat degradation. The introduction starts from very far away and brings the reader to a very narrow issue of interrelationship between two predators. I suggest the authors emphasize the effects of human presence and activity, as they also use it in the analyses, and limit the introduction to the description of effects of human presence on wildlife / carnivores and then lead to the hypothesis that different landscapes, with different degrees of human activities, may affect the presence and relative abundance of key predator species.
Study site: Not clear how the authors selected the locations to be investigated. The description of the landscapes is overly short, and the differences across landscapes in land use, climate, and human presence should be further elaborated. No indication of the size of each study site is provided, and how representative each one is of the landscape it belongs to. This section is poor and would benefit from a more elaborated figure.
Study design: It appears that the study design is opportunistic, as no indication is given on the selection of trap lines (grid nodes/centroids, or paths in suitable habitats?). The number of camera traps used appears to be low (13 in total), as is the number of days kept active. Furthermore, given that there is variation in number of trap/days and camera traps in each location, a discussion about the detectability of the different species is needed. See Bischof et al. (2014) for a discussion of variables affecting detectability of predators with camera traps, and Burton et al. (2015) for a discussion on the weaknesses when using camera traps for relative abundance studies. Finally, the use of an attractant for luring wildlife to camera traps could have an effect on the detectability of some individuals, as some may be attracted and others might avoid approaching the lured site, thus an unknown percentage of animals may never be caught by the camera trap. Thus, if the detectability of predators is usually improved using lures in presence/absence studies (Ferreras et al., 2018), the estimation of their abundance might be biased by the use of lures.
It is not clear how the authors have used images to obtain indexes of abundances of the target species only assuming a temporal independence set arbitrarily at 60 minutes. Such interval should be justified on the basis of behavioral characteristics of the targeted species. Abundance is usually estimated with capture-mark-recapture models that imply the capacity to individually identify the photographed animals. It is therefore very difficult to understand how the findings could be supported by the analyses and which assumptions were made.
Results: Table 1 could be an appendix. The variation between sites belonging to the same range should be better explained. The caption of table 4 should be revised as the codes are included in the caption of table 6.
The authors have missed to address some crucial points in their study design and the analyses should be revised to include detectability.